# Study of the Feasibility of Decoupling Temperature and Strain from a *ϕ*-PA-OFDR over an SMF Using Neural Networks

**DOI:** 10.3390/s23125515

**Published:** 2023-06-12

**Authors:** Andrés Pedraza, Daniel del Río, Víctor Bautista-Juzgado, Antonio Fernández-López, Ángel Sanz-Andrés

**Affiliations:** 1Instituto “Ignacio da Riva” (IDR), Universidad Politécnica de Madrid, 28040 Madrid, Spain; a.pedraza@upm.es (A.P.); antonio.fernandez.lopez@upm.es (A.F.-L.); 2ETSIAE, Universidad Politécnica de Madrid, 28040 Madrid, Spain; daniel.delrio.velilla@upm.es (D.d.R.); victor.bautista.juzgado@alumnos.upm.es (V.B.-J.)

**Keywords:** decoupling, distributed sensing, XAI, machine learning, *ϕ*-PA-OFDR

## Abstract

Despite several existing techniques for distributed sensing (temperature and strain) using standard Single-Mode optical Fiber (SMF), compensating or decoupling both effects is mandatory for many applications. Currently, most decoupling techniques require special optical fibers and are difficult to implement with high-spatial-resolution distributed techniques, such as OFDR. Therefore, this work’s objective is to study the feasibility of decoupling temperature and strain out of the readouts of a phase and polarization analyzer OFDR (ϕ-PA-OFDR) taken over an SMF. For this purpose, the readouts will be subjected to a study using several machine learning algorithms, among them Deep Neural Networks. The motivation that underlies this target is the current blockage in the widespread use of Fiber Optic Sensors in situations where both strain and temperature change, due to the coupled dependence of currently developed sensing methods. Instead of using other types of sensors or even other interrogation methods, the objective of this work is to analyze the available information in order to develop a sensing method capable of providing information about strain and temperature simultaneously.

## 1. Introduction

The general interest in Distributed Optical Fiber Sensors (DOFSs) has increased in the past few decades [1,2,3]. Due to the huge variety of phenomena related to light propagation along an optical fiber and depending on the interrogator and the fiber used, there are more than 60 types of sensors [1,2]. This technology has been applied in several fields such as Structural Health Monitoring [4], Geo-Hydrological applications [5], acoustic sensing [6] and even medicine [7].

Basically, a Distributed Optical Fiber Sensor consists of a segment of optical fiber within which monochromatic light is propagated forward (transmission) and backward (backscattering). The fiber core’s shape is varied, going from a simple circle up to the polarization-maintaining configurations shown in Figure 1. The dimension of this core determines the number of modes that can be propagated. In the present work, light propagation into an SMF is studied. This type of fiber is preferred due to its lower cost and reduced size when compared with other types of optical fiber (PMF), where, moreover, temperature-strain decoupling has been currently achieved.

Commonly, light propagated within the fiber is emitted by a tunable laser source; and the backscatter radiation is analyzed by a photodetector after passing through some filters and beamsplitters [8]. Both photonic systems are usually integrated into the same equipment, and the emitted and analyzed light wavelength determines the type of interrogator (Raman, Rayleigh, or Brillouin [3]), and with the appropriate beamsplitters and filters, both states of polarization can be studied separately. The phase analysis needs a special configuration in the photodetector to compare the signal received with a local oscillator [9]. In the present work, a Phase-sensitive and Polarization Analyzer Optical Frequency Domain Reflectometer (ϕ-PA-OFDR) was used.

Light backscattering is sensitive to both temperature and strain changes and is presently the principal limitation when implementing this technology. The solutions are diverse and depend on the type of DOFS chosen, but two main groups can be distinguished. The first type of solution is to integrate sensors in parallel, which are usually either strain gauges or thermocouples along with Optical Fiber Sensors (OFSs) in order to obtain one independent measure. The second type of solution consists of a signal treatment, which requires either a second type of interrogator or a signal post-processing. In Table 1, some of the methods developed for different OFSs are listed.

More specifically, the last one is the most similar to the present work. In it [27], Froggat et al. use a ϕ-PA-OFDR over a PMF. However, in the present work, the fiber studied is an SMF because of its higher availability and lower price (around two orders of magnitude [28,29]). The behavior of states of polarization in an SMF is different, and the same technique used by Froggat et al. cannot be implemented. More specifically, the birefringence vector’s randomness causes fast and slow polarization modes to have different time arrivals. In addition, they will decompose into both the fast and slow modes of the next segment, leading to polarization-mode coupling [30].

Artificial intelligence has been applied successfully to Optical Fiber Sensors [31]. Therefore, before trying to find an analytical approach to the states of polarization evolution through the fiber or any statistical analysis of the signals acquired, artificial intelligence (AI) methods have been applied to the problem in the present work, in order to determine the feasibility of decoupling temperature and strain from a ϕ-PA-OFDR readout.

## 2. Materials and Methods

### 2.1. Resources Used

The optical fiber and interrogator used are described in Table 2. Moreover, the signal was analyzed using Python open source libraries as Scipy [32] and Numpy [33].

### 2.2. Experimental Data Acquisition

For a proper creation of an artificial intelligence model, it is imperative to have a huge and reliable experimental dataset [37,38]. Therefore, the optical fiber was subjected to mechano-thermal tests where the longitudinal fiber strain and temperature were varied. For this purpose, a cantilever aluminum beam is supported at one end, restricting all six degrees of freedom, and some weights are hanging at the free end to produce a linear strain field on the top face of the beam, where the SMF is fixed with cyanoacrylate glue. This solution is taken instead of setting the deflection of the beam with some mechanism because of the temperature variation. Then, the cantilever beam is placed in a calibrated stove to set different temperatures to acquire different sets of data. Finally, SMF was converted into a succession of overlapping sensors with a sensor length of 20 mm and a sensor spacing of 2 mm. This is possible because of the interferometer used, which provides 2000 samples per sensor, which allows signal processing calculations.

In Figure 2 the experimental setup is illustrated, as well as the temperature and strain distribution shape and how the fiber readouts along the beam are converted into a row of sensors.

### 2.3. Interferometer Readouts

A ϕ-PA-OFDR provides information about the amplitude and phase of the two states of polarization of the interference between the backscattered electromagnetic field $E_{m\;}$ and the one generated by a local oscillator $E_{lo}$ (see Figure 3 or [8] for more details). This means that two complex measurements, *S* and *P*, are received by the photo-detectors every time the optical fiber is interrogated,
(1)S(ω)=2ReEloT¯¯s†T¯¯sr¯¯(ω)Emexp[jω(t)Δτ],P(ω)=2ReEloT¯¯p†T¯¯pr¯¯(ω)Emexp[jω(t)Δτ],
where the matrix operators ${\bar{\bar{T}}}_{s}$ and ${\bar{\bar{T}}}_{p}$ represent the beam splitter action, $\bar{\bar{r}}\left(\omega \right)$ is the complex spectral reflectivity, and Δτ is the temporal delay. Once the spectral response is obtained, the time-domain response of the sensing fiber can be obtained by means of a Discrete Inverse Fourier Transform (DIFT). Then, knowing the speed of light in the fiber, obtaining the spatial distribution is trivial.

## 3. Results

### 3.1. Dataset

The process carried out to create the dataset is illustrated in Figure 4. The readings obtained from the test under controlled conditions have been labeled with the test conditions themselves. Then each of these signals has been taken and divided into segments that act as a point sensor; i.e., they can be assigned a single temperature and a single strain value. Finally, these segments have been taken two-by-two and the temperature and strain increment corresponding to the pair has been computed, thus storing the two segment signals (*P* and *S*) and the difference between states. As the most suitable pre-treatment of the signals was not known a priori, it was decided to keep both signals ($P_{1},S_{1},P_{2},S_{2}$) in full in order to evaluate the performance of the different transformations.

From the test performed, data have been obtained at four temperatures, 20, 30, 40, and 50 ${}^{\circ }$C, and five strain states associated with an end deflection of, 0, 3.11, 6.22, 9.33, and 12.45 mm. These combined states result in 20 different states, which, combined two-by-two, amount to 190 possible combinations. By means of the optical interrogator used, the optical fiber was converted into a succession of overlapping sensors with a sensor length of 20 mm (with a sampling period of 10 μm) and a sensor spacing of 2 mm (see Figure 2). Then, a fiber length of 300 mm, of which only measurements between 100 and 280 mm have been selected (to avoid possible errors), consists of a total of 80 sensors. Randomly taking a percentage of the number of combinations, a dataset of 13,950 samples has been created by taking, randomly again, 60% for training, 20% for validation, and the remaining 20% for testing.

The temperature range has been determined by the room temperature at the time of the test (since the oven does not have refrigeration) and the characteristics of the adhesive used, cyanoacrylate, to adhere the optical fiber to the aluminum plate.

### 3.2. Pre-Processing

Sometimes, Artificial Intelligence is presented as a tool capable of solving any problem with just the corresponding training time. Nothing could be further from the truth: the proper dataset pre-processing is key not only in the success or failure of the model but also in its accuracy and efficiency [39,40].

In the beginning, the use of the raw *P* and *S* signals was attempted with poor results. Then, the correlation between signals was proposed as a possible pre-processing due to its presence in the relative spectral shift calculation process, but the results of the non-supervised algorithms were still poor. Since Froggat et al. [27], used the auto-correlation to achieve the temperature–strain decoupling, the autocorrelation of the signals was also included. Just then, the machine learning algorithms began to show promising results. Finally, the following correlation between signals was chosen:$$X=\left[P_{1}🞰P_{1},P_{1}🞰P_{2},P_{2}🞰P_{2},S_{1}🞰S_{1},S_{1}🞰S_{2},S_{2}🞰S_{2}\right]\;,$$
where the six signals were concatenated in a one-dimensional array. The process is illustrated in Figure 5.

### 3.3. Clustering

Before starting to apply machine learning models, a clustering algorithm was used to determine if the pre-processed data were capable of being classified into groups that somehow could be related to strain and temperature increments. Note that this is an iterative process where the dataset was exposed to different pre-processing methods; however, only the final results of the clustering algorithm are shown in Figure 6. As can be seen, the algorithm (TSNE) spontaneously generated temperature groups and orders the deformations within them, which means that although there is some confusion within some groups, an artificial intelligence model is able to differentiate both variables.

### 3.4. Neural Network


**Input data**


Once this check has been carried out, the artificial intelligence model capable of discerning between the temperature increase and deformation from the signals of the different polarizations is designed. After trying several options to train the model, an input vector, composed of the cross-correlation of the two polarization states and the four auto-correlations of the four available measurements, is selected due to the clustering algorithm results. The input to the network is finally as follows:$$x_{j}=\left[P_{1}🞰P_{1},P_{1}🞰P_{2},P_{2}🞰P_{2},S_{1}🞰S_{1},S_{1}🞰S_{2},S_{2}🞰S_{2},\Delta \nu \right]\;,$$
where the frequency increment Δν has been added to provide scaling information (since the same equipment can operate in different frequency ranges).


**Normalization**


These input vectors have been previously normalized since training is much more efficient with normalized values. In this case, the normalization of each variable separately has been finally selected as it is the only one with which the model was able to fit the data:(2)$$\begin{array}{cc} X= & \left(\begin{array}{c} x_{1}\\ x_{2}\\ \cdots \\ x_{n_{s}-1}\\ x_{n_{s}} \end{array}\right)\to \begin{array}{c} X_{\max\;i}=\max\left(\{X_{0i},\cdots ,X_{ji},\cdots X_{n_{s}i}\}\right)\\ X_{\min\;i}=\min\left(\{X_{0i},\cdots ,X_{ji},\cdots X_{n_{s}i}\}\right) \end{array}\to \\ \; & \;\to {\hat{X}}_{ji}=\frac{X_{ji}-X_{\min\;i}}{X_{\max\;i}-X_{\min\;i}}\;, \end{array}$$
where index *i* refers to the column index, that is, to each of the items that make up the vector $x_{j}$; on the other hand, index *j* refers to the sample number.


**Architecture**


The network architecture is as shown in Figure 7 and is basically a compendium of densely connected layers with hyperbolic tangent-type activation functions that add to its nonlinearity. The architecture chosen consists of two stages, one for the “feature extraction” and the other for the “regression” of each magnitude (strain and temperature) with more layers to make the dimensional reduction smoother. This simple architecture was capable of regressing the values in an acceptable range so more complex models such as Convolutional Neural Networks or Generative Adversary Neural Networks were not developed. However, for future work, these architectures will be tested.


**Training**


The results of the network training are shown in Figure 8. The training has been carried out with an Adam-type optimizer with a learning rate of $10^{-6}$ and computing the error with the mean squared error (MSE) criterion.

As can be seen at the 100 training epochs, the behavior is asymptotic, which indicates that the model is not able to fit more of the data. On the other hand, from the behavior of the validation curves, it can be determined that the model has not undergone overfitting. If the model were remembering data, the behavior of training and validation losses would not be asymptotic; instead, the validation curve would be enlarging as long as the model is only remembering past samples and the validation samples (never used for train) are completely unknown for the model.


**Results**


Once the model has been trained, it is evaluated with the test data. In Figure 9 and Figure 10, error histograms for the target variables are shown together with an approximation of normal distribution, whose coefficients are shown in Table 3. In addition, confidence intervals of 99% and 95% were computed, and the limits of these intervals are also shown in Table 3.

For better comprehension, the predicted versus target values have also been plotted on a plane (see Figure 11 and Figure 12) to create the equivalent of a confusion matrix but with continuous data. If the model worked perfectly, the predicted and target data would be the same, thus plotting a diagonal. For this reason, a least squares regression line has been plotted over the plane. The regression line has the form y=mx+n, where *x* represents the target value and *y* the predicted one, *m* the slope of the straight line, and *n* the ordinate at the origin. Additionally, the value of $r^{2}$ has also been computed to determine the dispersion of the values. All of the above values can be also found in Table 3.


**Explainable Artificial Intelligence (XAI)**


Explainable Artificial Intelligence (XAI) consists of a series of methods aimed at converting the results provided by artificial intelligence into a form that humans can understand. It contrasts with the “black box” concept of machine learning, where even its designers cannot explain why an AI arrived at a particular decision. Thanks to XAI methods, features can be extracted that allow existing knowledge to be confirmed, existing knowledge to be questioned, and new hypotheses to be generated.

In this case, XAI allows explaining how the developed model interprets the correlations of the signals to reveal the information on which the actions are based. To implement the XAI methods on this model, the Lime-For-Time repository (see [41]) has been used, in which the LIME library ([42]) is used to analyze time series.

The analysis consists of taking an example signal and dividing it into segments, in this case 120, given the composition of the signal. Next, the neural network is studied as a classifier in which each segment constitutes a class and which contributes, to a greater or lesser extent, to the final decision. From the example signal, multiple variations in the parameters are made to interpret how the model behaves to the different inputs. From these data, the relevance of each of the segments can be determined. Taking the 12 most relevant segments and displaying their weights in a histogram, the images shown in Figure 13 and Figure 14 are produced.

If these segments are displayed over the signal, it can be seen which regions are the most relevant. In this case, to represent the importance, an opacity has been assigned according to the weights previously shown in the histograms. The result of this graphical representation is the one shown in Figure 15 and Figure 16.

## 4. Discussion

From the results shown in the previous section (Section 3.4), it can be said that an artificial intelligence capable of distinguishing between strain and temperature readings has been trained successfully, and so, the readout of the ϕ-PA-OFDR contains enough information to obtain the measure of the two magnitudes (strain and temperature) even when an SMF is used for sensing.

The results show how the artificial intelligence recognizes the spectral shift by comparing the values of segments 88, 30, and 89, which correspond to the central area of the cross-correlations between the two compared signals. On the other hand, studying the rest of the values is interesting, as this is where the ability to discern between thermal and mechanical effects lies.

Comparing the two images in Figure 15 and Figure 16, it can be seen how, to determine the temperature, the model takes two symmetrical regions of the autocorrelation of the polarization state P of the first signal and similar regions of the autocorrelation of the polarization state S of the second one. In addition, to determine the deformations, it is observed that the last values of the autocorrelation of the polarization state S of the second signal are interpreted by some kind of integral.

Additionally, the temperature-related decision seems to focus more on the behavior of the peak of the autocorrelation of the P polarization state of the second signal, whereas the deformations focus on the peak of the autocorrelation of the S polarization state of the first one.

This information allows us to determine that the autocorrelation of the signals does provide additional information that can be key in determining unequivocally at what temperature and state of deformation the optical fiber is. However, it should be noted that the results provided by the AI may not follow any logic because the algorithm obviously does not know anything about fiber optic sensing.

## 5. Conclusions

From these results, the following conclusions can be drawn:Readouts provided by ϕ-PA-OFDR (as OBR-4800 is) contents are capable of providing more information than what is presently used.Artificial intelligence methods are suitable for analyzing DOFS data in order to decouple temperature and strain phenomena.More specifically, the the neural network model designed and trained for this purpose in the present work has reached the precision and accuracy shown in Table 4.In addition, explainable AI offers a deeper analysis of the AI model, which can be used to chart the course of future research.

## Figures and Tables

**Figure 1 sensors-23-05515-f001:**
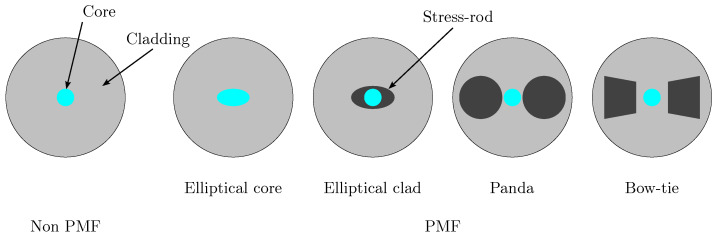
Different existing core shapes for optical fiber. On the **left**, a cross-section of a Non-Polarization-Maintaining Optical Fiber (non PMF) with a circular core. On the **right**, a cross-section of four types of PMF with different types of core or stress-rods into the cladding.

**Figure 2 sensors-23-05515-f002:**
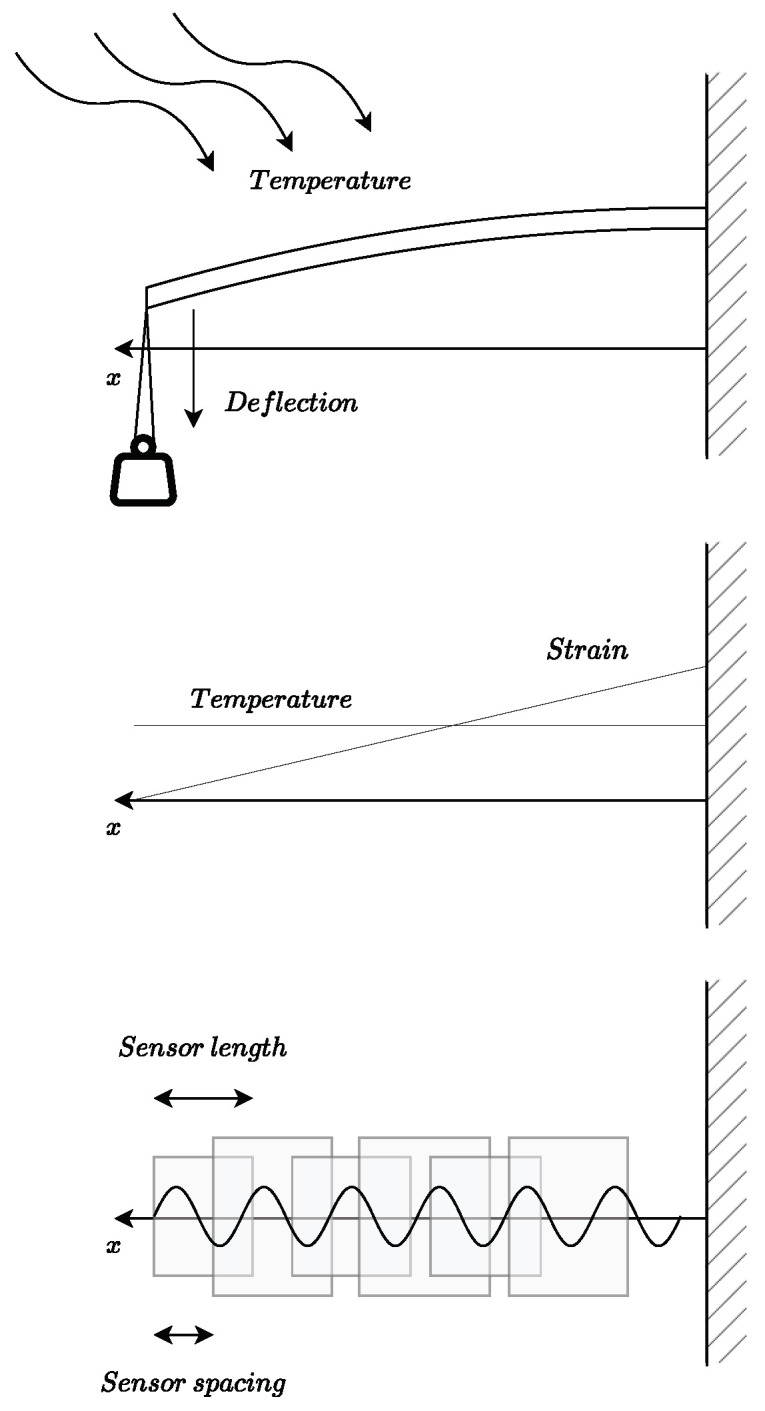
(**Top**): Test configuration, a cantilever aluminum beam (a plate of 300 × 30 × 2 mm with a Young modulus of E = 70.3 GPa and a Coefficient of Thermal Expansion of α = 24 μm/(m·K)) fixed at one end and up to four 50 g weights added at the free end (since the weight is maintained with temperature). (**Middle**): Strain and temperature distributions for any test. (**Bottom**): A scheme of the spatial distribution of the point sensors along the beam length.

**Figure 3 sensors-23-05515-f003:**
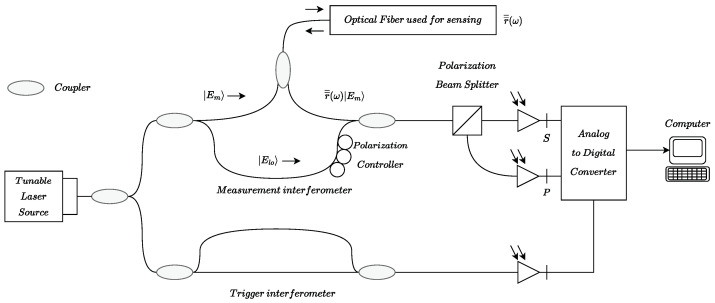
Measurement network for a Polarization Analyzer OFDR. In [8], Soller et al. describe this measurement network more precisely.

**Figure 4 sensors-23-05515-f004:**
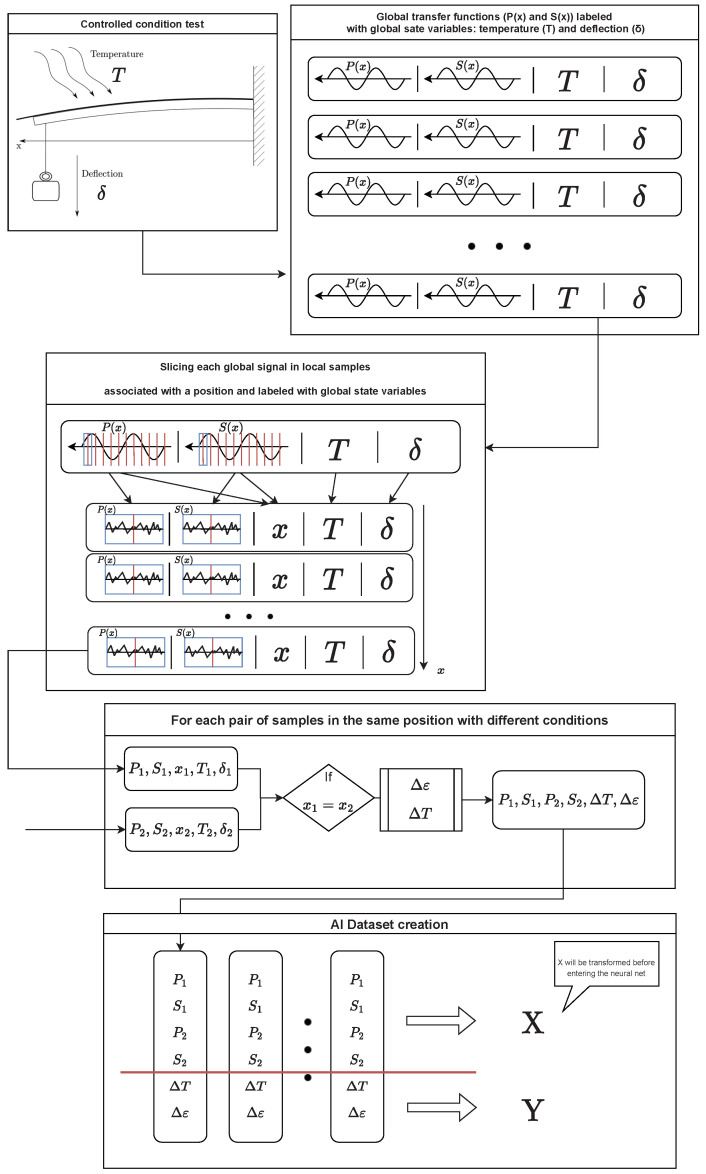
Process carried out to obtain the dataset with which the model has been trained. Firstly, the acquired readouts are labeled according to their conditions of temperature and deflection at the free end of the beam. Then, each readout is sliced into several samples, where it is assumed that the strain and temperature are constant. After, for each pair of slices with the same spatial position, the temperature and strain increment are computed. Finally, this set of data is composed of two slices of readouts with two polarization states (four signals), and their corresponding temperature and strain increments are appended to the dataset.

**Figure 5 sensors-23-05515-f005:**
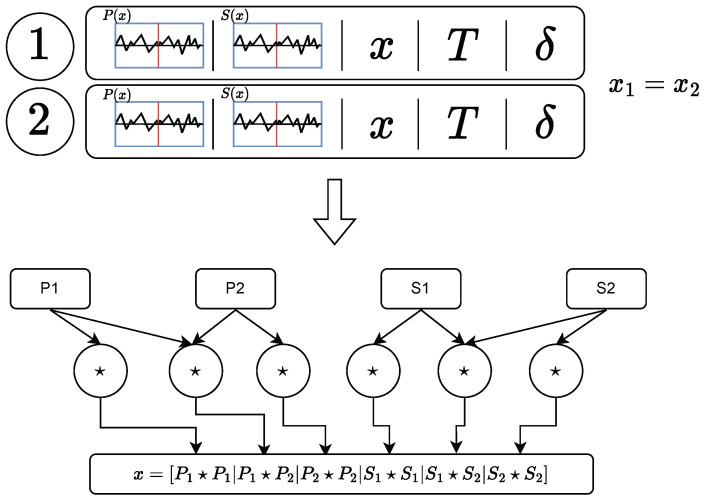
Process of readout preprocessing: two slices of the full readout (labeled with ”1” and ”2”) that matches in position are taken, then its signals are cross-correlated or auto-correlated (operation denoted with a star 🞰), and the six corresponding signals are concatenated in one array.

**Figure 6 sensors-23-05515-f006:**
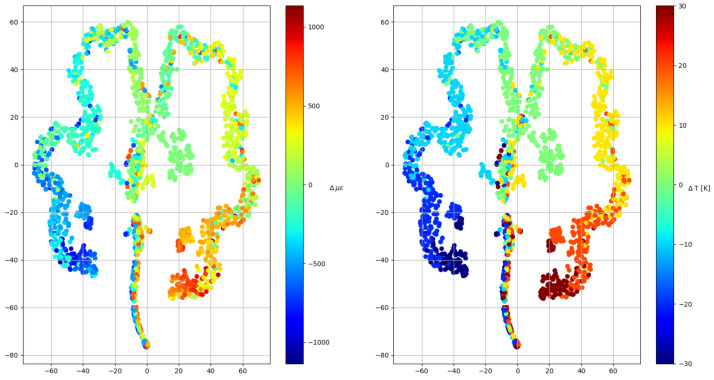
Clustering algorithm results. The algorithm takes a vector with N dimensions and converts it into 2 dimensions, so that all the samples are represented in a plane. On the left, the samples are colored according to their strain increment and at right according to their temperature increment.

**Figure 7 sensors-23-05515-f007:**
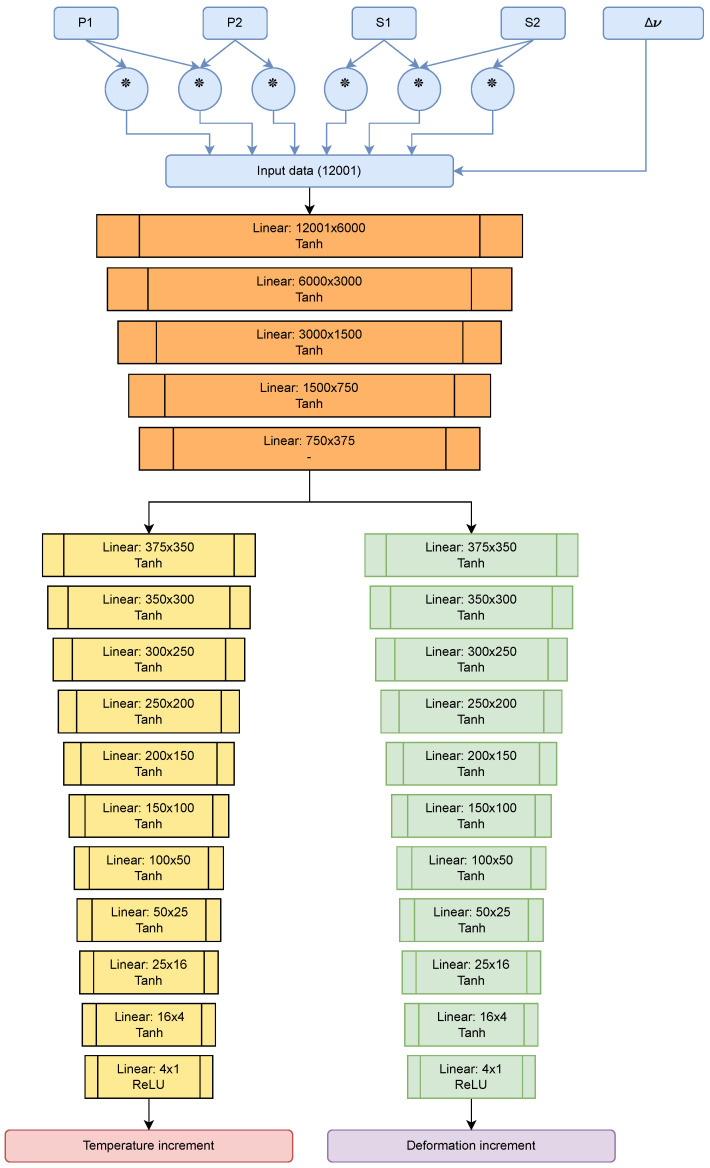
Diagram of the neural network used. In the upper part, in blue, the pre-processing of the signal prior to the neural network is shown; the operation marked with the ∗ denotes correlation between two input signals.

**Figure 8 sensors-23-05515-f008:**
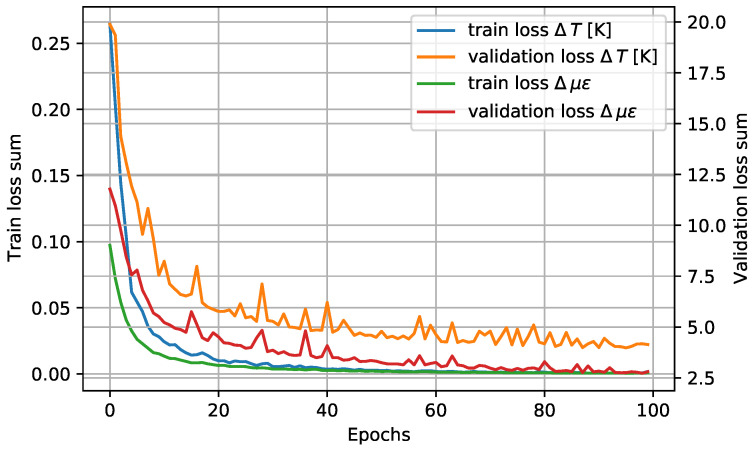
Training and validation losses of the network for both target variables.

**Figure 9 sensors-23-05515-f009:**
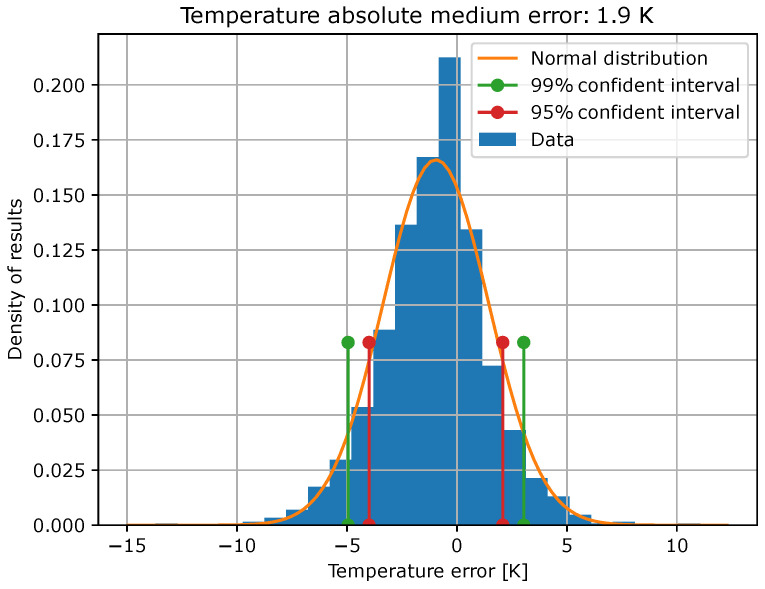
Histogram of temperature errors and their normal distributions, as well as its confidence interval limits.

**Figure 10 sensors-23-05515-f010:**
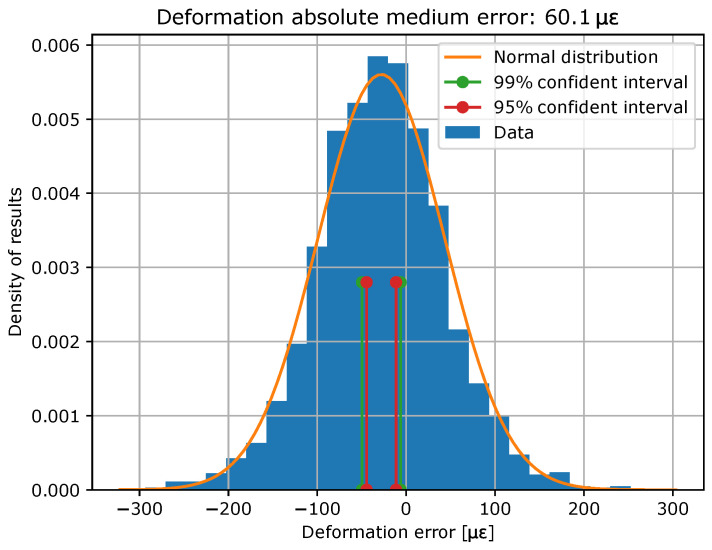
Histogram of strain errors and their normal distributions, as well as its confidence interval limits.

**Figure 11 sensors-23-05515-f011:**
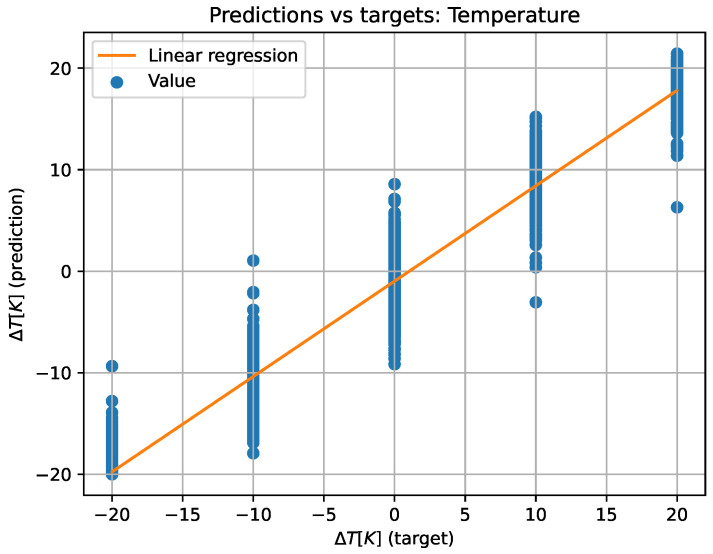
Two-dimensional representation of predicted versus target values and linear regression for temperature data.

**Figure 12 sensors-23-05515-f012:**
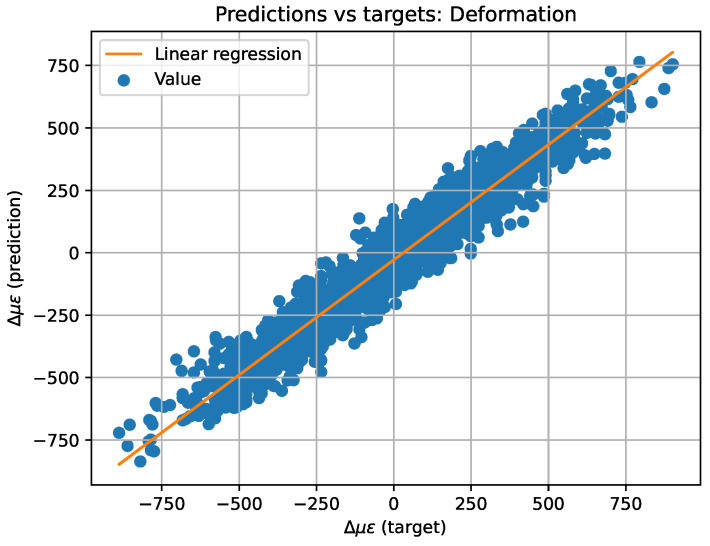
Two-dimensional representation of predicted versus target values and linear regression for strain data.

**Figure 13 sensors-23-05515-f013:**
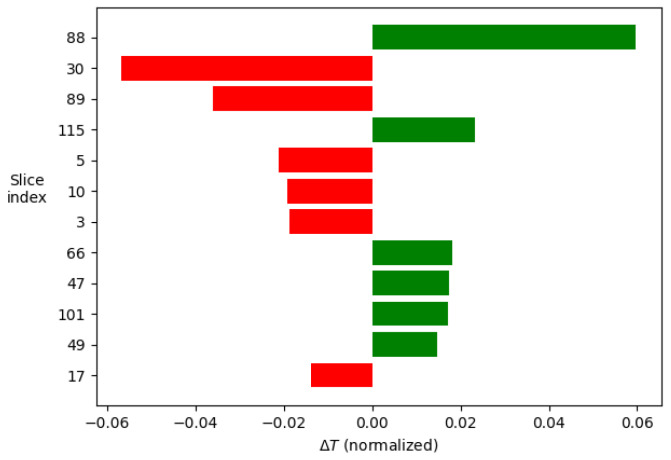
Weight in the final decision of each of the segments of the signal in the order of relevance for the temperature decision.

**Figure 14 sensors-23-05515-f014:**
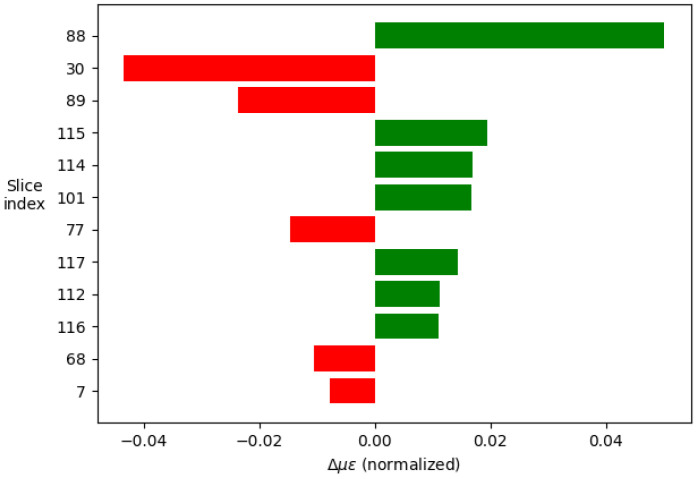
Weight in the final decision of each of the segments of the signal in the order of relevance for the strain decision.

**Figure 15 sensors-23-05515-f015:**
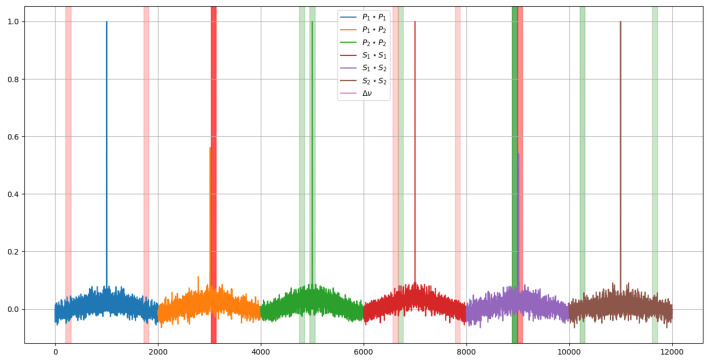
Most relevant regions of the signal for each of the model predictions for the temperature prediction.

**Figure 16 sensors-23-05515-f016:**
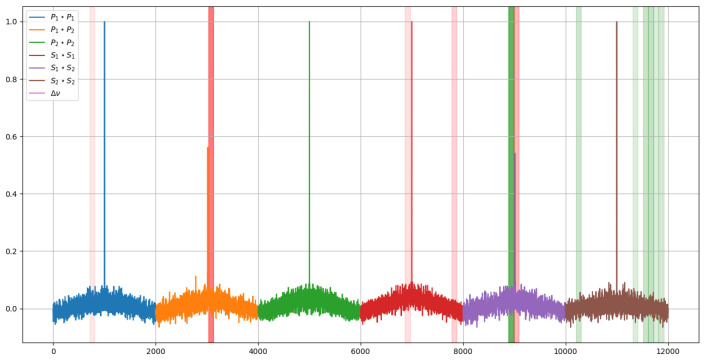
Mostrelevant regions of the signal for each of the model predictions for the strain prediction.

**Table 1 sensors-23-05515-t001:** A summary of the strain/temperature discrimination methods using DOFS configurations.

Method	Configuration	Refs.
Brillouin	Brillouin frequency shift	[10,11]
	Brillouin amplitude effects with LEAF fiber	[12,13]
	Dispersion shifted fiber	[14]
Brillouin and Raman hybrid	Raman–Brillouin gains	[15,16,17]
Brillouin and Rayleight hybrid	Frequency shift	[18]
Rayleight scattering	Fiber with core-offset splicing	[19]
Photonic crystal fiber (PCF)	Brillouin	[11,20,21,22,23]
FBG	Dual wavelength	[24]
FBG on PMF	Polarization-Maintaining Fiber analysis	[25]
OTDR-BOTDA	Time delay	[26]
OFDR	PMF, Rayleigh scattering, auto-correlation function	[27]

**Table 2 sensors-23-05515-t002:** Resources used.

Resource	Supplier	Model	References
Single-Mode Fiber	Corning (Corning, NY, USA)	SMF-28e+	[34]
Optical Frequency Domain Reflectometer	LUNA (Roanoke, VA, USA)	OBR-4600	[35,36]

**Table 3 sensors-23-05515-t003:** Deep Learning AI model error metrics.

	Normal Distribution		Confident Intervals		Linear Regression		
	μ	$\sigma^{2}$	99%	95%	*m*	*n*	$r^{2}$
ΔT [K]	−0.5948	2.4044	[−4.5889,3.3994]	[−3.6339,2.4444]	0.94	−0.61	0.96
Δμε	−27.2083	71.4478	[−48.9810,−5.4357]	[−43.7753,−10.6414]	0.93	−27.39	0.94

**Table 4 sensors-23-05515-t004:** Deep Learning AI model error metrics.

	Normal Distribution		Confident Intervals		Linear Regression		
	μ	$\sigma^{2}$	99%	95%	*m*	*n*	$r^{2}$
ΔT [K]	−0.5948	2.4044	[−4.5889,3.3994]	[−3.6339,2.4444]	0.94	−0.61	0.96
Δμε	−27.2083	71.4478	[−48.9810,−5.4357]	[−43.7753,−10.6414]	0.93	−27.39	0.94

## Data Availability

Not applicable.

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
