# Peer review of "Study of the Feasibility of Decoupling Temperature and Strain from a ϕ-PA-OFDR over an SMF Using Neural Networks"

_sensors, 2023, doi:10.3390/s23125515_

Round 1

Reviewer 1 Report

I read the manuscript with interest; frankly, all the work on the application of artificial intelligence tools to one-dimensional signal processing is quite interesting for me. The approach described in this paper is very appealing, especially since it touches on aspects of Explainable Artificial Intelligence tools. This sets this paper apart from other works on similar topics. I will certainly recommend the manuscript for publication after the authors have made minor revisions and answered my questions. 

1. I would like to draw the authors' attention to the order in which the literature is cited. The references should be numbered as they are mentioned. 

2. Section "2.3. Interferometer readouts", including formulas and figure, completely repeats parts of the article [12]. How valuable is the duplication of material in this article? The formulas and survey diagram given are not relevant to the topic of the manuscript per se and do not add value to it.

3. The abbreviation SOP is only used 4 times in the text in lines (34, 49, 55, and 72). I would recommend using the full term to make the text read better. The abbreviation AI (Artificial Intelligence) is deciphered twice in lines 56 and 167.

4. On lines 109-110, the authors write that they considered various options for preprocessing signals, but eventually chose a correlation of the form

X = [P1 ⋆ P1, P1 ⋆ P2, P2 ⋆ P2, S1 ⋆ S1, S1 ⋆ S2, S2 ⋆ S2].

I would like to know what other options were considered and why we settled on it. What does the functional dependence of this correlation (or what is fed to the input of the neural network) look like?

5. On lines 141-143, the authors briefly describe the chosen neural network architecture. I would like to know why they chose such a cumbersome and complex network to solve this seemingly simple problem. It seems to me that this architecture has too many parameters, It leads to "remembering" the data rather than approximating it.

Reviewer 2 Report

I believe that the authors of the presented work have chosen a very timely topic. Their study is relevant, has a pronounced scientific novelty and will be of interest to the readers of the journal. However, I think that there are some shortcomings in the paper that should be corrected before the publication in Sensors.

- Line 4: Please let the readed know, what is PA;
- I have to note that the literature review part in the paper is rather poor. Unfortunately, the authors talk about the OFDR/OBR technology, presenting it as the same for all types of fiber optic sensors. In the introduction, it would be correct to talk about distributed fiber-optic sensors based on Brillouin and Raman scattering too, to pay attention to various approaches: reflectometry in both the time and frequency domains.
Yes, the authors provide Table 1, but all their Refs are related to OFDRs and there's no another method to compare with. To improve the scientific background part related to other state-of-the-art techniques, I propose to mention the following studies: single-shot hybrid CP-φOTDR/CP-BOTDA system for simultaneous distributed temperature/strain sensing [10.1364/OFC.2022.Th2A.15]; temperature compensation technology of BOTDR strain monitoring [10.1115/ISPS2013-2867]; simultaneous measurement of optical fibre deformation and temperature in a hybrid distributed sensor based on the detection of Rayleigh and Raman scattering [10.1070/QEL16541]; theoretical and experimental estimation of the accuracy in simultaneous distributed measurements of temperatures and strains in anisotropic optical fibers using polarization-Brillouin reflectometry [10.1134/S0020441220040223]; highly sensitive curvature sensor based on single-Mode fiber using core-offset splicing [10.1016/j.optlastec.2013.09.036];
- Line 16: The Refs numeration looks randomly in the entire manuscript, why do we start from the Ref [9]?;
- Line 17: 'here are more than 60 types of sensors [3]'. Are all them in Ref [3]? Or OFDR/OBR-types only?
- Line 77: 'By means of a Discrete Inverse Fourier Transform (DIFT) the time-domain response of the Device Under Test (DUT) can be obtained' - are you sure it's really inversed? Following the works by Soller, Gifford et al, they use the forward DFT for the entire signal, but the DIFT was appled in the scanning window on the next stages.
- In addition, the entire signal preprocessing procedure is not described clearly enough. The experiment must be repeatable. For example, it is known that information about temperatures and strains is contained in the phase spectrum, but not in the amplitude spectrum (which, in fact, is a typical OFDR-trace, displayed by default by OBR). This process is very clearly described in [10.1364/AO.57.001424]. Is this possible to do it some similar way?
- Line 78: 'in' instead of 'into';
- Figure 3 should be redrawn since it is copied from the cited Ref;
- Why the neyral network architecture shown in Figure 6 was chosen? It should be explained in the text;
- The captions for many figures, such as Figure 5, are overly laconic. It is necessary that for all graphs a detailed description in the text be provided, and the captions contain the names of physical quantities and units of measurement displayed on the axes. In addition, some sections of the article end with Figures. I recommend moving them higher, immediately after the first mention in the text.

The English is quite OK but some typos need to be fixed

Reviewer 3 Report

  1. Currently, most distributed fiber optic sensors utilize scattering in the optical fiber for strain sensing, with both Brillouin scattering and Rayleigh scattering being temperature-sensitive. Therefore, the cross-sensitivity issue between temperature and strain is a very common topic in distributed fiber optic sensors. This paper attempts to decouple temperature and strain by employing machine learning and deep learning methods on φ-PA-OFDR, which is a promising approach. But I think there are many issues to be tackle before the paper is published. 1. This article is not well-written and lacks coherent logic. For instance, the concept of φ-PA-OFDR is not adequately described or precisely defined. The order of the references is quite disorganized and should be adjusted. 2. What does it mean to achieve decoupling with single-mode fibers in line 28? 3. I am unsure of the purpose behind including the cross-sectional diagram of different types of optical fibers in line 29, and it seems unnecessary. 4. In line 54, the author mentions that artificial intelligence techniques have been applied in optical fiber sensors (OFS), so they can be used to analyze SOP (state of polarization) and solve the decoupling problem between temperature and strain. However, this argument does not hold logically. 5. The explanation of the experimental setup is too brief, and it is difficult to fully understand how the experiment was conducted from Figure 2. 6. The author states, 'the temperature-related decision seems to focus more on the behavior of the peak of the autocorrelation of the P polarization state of the second signal, whereas the deformations focus on the peak of the autocorrelation of the S polarization state of the first one.' It would be better to interpret this phenomenon from the perspective of fiber optic sensing, as I find this explanation illogical.

Round 2

Reviewer 2 Report

I thank the authors for their careful work. I believe that the paper is now ready for the publication in Sensors.

Reviewer 3 Report

The author provided well responses to the reviews. It is recommended to proceed with the publication.